# Bone marrow stromal cells induce an ALDH+ stem cell-like phenotype and enhance therapy resistance in AML through a TGF-β-p38-ALDH2 pathway

Bin Yuan[1]ᵒ, Fouad El Dana[1]ᵒ, Stanley Ly[1], Yuanqing Yan[2], Vivian Ruvolo[1], Elizabeth J. Shpall[3], Marina Konopleva[1], Michael Andreeff[1]*, Venkata Lokesh Battula[1]*

**1** Department of Leukemia, Section of Molecular Hematology and Therapy, The University of Texas MD Anderson Cancer Center, Houston, Texas, United States of America, **2** Department of Neurosurgery, The University of Texas Health Science Center at Houston, Houston, Texas, United States of America, **3** Department of Stem Cell Transplantation and Cellular Therapy, The University of Texas MD Anderson Cancer Center, Houston, Texas, United States of America

ᵒ These authors contributed equally to this work.
* mandreef@mdanderson.org (MA); vbattula@mdanderson.org (VLB)

**Data Availability Statement:** All RNA sequencing data files are available from the NCBI database (accession number GSE152996). All other relevant

## Abstract

The bone marrow microenvironment (BME) in acute myeloid leukemia (AML) consists of various cell types that support the growth of AML cells and protect them from chemotherapy. Mesenchymal stromal cells (MSCs) in the BME have been shown to contribute immensely to leukemogenesis and chemotherapy resistance in AML cells. However, the mechanism of stroma-induced chemotherapy resistance is not known. Here, we hypothesized that stromal cells promote a stem-like phenotype in AML cells, thereby inducing tumorigenecity and therapy resistance. To test our hypothesis, we co-cultured AML cell lines and patient samples with BM-derived MSCs and determined aldehyde dehydrogenase (ALDH) activity and performed gene expression profiling by RNA sequencing. We found that the percentage of ALDH+ cells increased dramatically when AML cells were co-cultured with MSCs. However, among the 19 ALDH isoforms, ALDH2 and ALDH1L2 were the only two that were significantly upregulated in AML cells co-cultured with stromal cells compared to cells cultured alone. Mechanistic studies revealed that the transforming growth factor-β1 (TGF-β1)-regulated gene signature is activated in AML cells co-cultured with MSCs. Knockdown of TGF-β1 in BM-MSCs inhibited stroma-induced ALDH activity and ALDH2 expression in AML cells, whereas treatment with recombinant TGF-β1 induced the ALDH+ phenotype in AML cells. We also found that TGF-β1-induced ALDH2 expression in AML cells is mediated by the non-canonical pathway through the activation of p38. Interestingly, inhibition of ALDH2 with diadzin and CVT-10216 significantly inhibited MSC-induced ALDH activity in AML cells and sensitized them to chemotherapy, even in the presence of MSCs. Collectively, BM stroma induces ALDH2 activity in AML cells through the non-canonical TGF-β pathway. Inhibition of ALDH2 sensitizes AML cells to chemotherapy.

data are within the paper and its Supporting Information files.

**Funding:** This work is supported by grants from the National Institute of Health (CA100632), Institutional Research Grant (IRG) from MD Anderson Cancer Center, Cure Sonia Foundation and Golfer's Against Cancer to VLB. In addition, this work was supported by grants from the National Institutes of Health (CA055164) and the MD Anderson Cancer Center Support Grant (CA016672), Cancer Prevention Research Institute of Texas (CPRIT, RP121010), and the Paul and Mary Haas Chair in Genetics to MA.

**Competing interests:** The authors have declared that no competing interests exist.

**Abbreviations:** ALDH, Aldehyde dehydrogenase; AML, Acute myeloid leukemia; ANOVA, Analysis of variance; BM, Bone marrow; BME, Bone marrow microenvironment; BM-MSC, Bone marrow-derived mesenchymal stromal cells; FACS, Fluorescence-activated cell sorting; FBS, Fetal bovine serum; MSC, Mesenchymal stromal cells; PBS, Phosphate-buffered saline; PBS-T, 0.05% Tween 20 in phosphate-buffered saline; RPMI, Roswell park memorial institute medium; TGF-β, Transforming growth factor-β.

## Introduction

The bone marrow microenvironment (BME) contributes to acute myeloid leukemia (AML) growth and chemotherapy resistance. The interaction between AML cells and other cells in the BME provides crucial support through the secretion of growth factors and chemokines, accelerating tumor growth and interfering with chemotherapy delivery [1, 2]. Mesenchymal stromal cells (MSCs) in the bone marrow (BM) are critical for growth induction and anti-apoptotic signaling in AML [3]. However, the mechanisms of stroma-mediated AML growth and chemotherapy resistance are not clear. As leukemogenesis and chemotherapy resistance are characteristics of AML stem cells, we hypothesized that the BME induces a stem cell-like phenotype in AML cells.

Several signaling pathways contribute to chemotherapy resistance in AML through induction of a high-mesenchymal, stem-like cell state [4]. Among them, transforming growth factor-β (TGF-β)-mediated canonical and non-canonical pathways have been well characterized in AML cells. TGF-β binds to its receptor, which phosphorylates Smad2 and Smad3. These proteins form a complex with Smad4, translocate to the nucleus, and regulate the expression of target genes in close association with several transcription factors. This is known as the TGF-β canonical signaling pathway. TGF-β can also activate multiple other pathways independently of Smads. Through the non-canonical pathway, TGF-β induces activation of Erk through Ras, Raf, and their downstream MAPK cascades, including p38 MAPK, which phosphorylates p38. Erk then regulates target gene transcription through its downstream transcription factors [5–7]. Inhibition of TGF-β signaling using small molecule inhibitors or receptor-blocking antibodies inhibited leukemia growth and sensitized AML cells to chemotherapy [5]. TGF-β signaling has cell type–specific effects and has been involved in the induction of a stem cell–like phenotype in solid tumors [8–11].

Aldehyde dehydrogenase (ALDH) is an enzyme involved in oxidizing toxic aldehydes into neutral acids [12]. ALDH activity is increased in hematopoietic stem cells and leukemia stem cells. In fact, there is an assay developed using ALDEFLUOR® to prospectively isolate hematopoietic stem cells and leukemic stem cells from peripheral blood or bone marrow samples [13, 14]. ALDH-positive (ALDH+) leukemia cells have greater tumorigenicity and chemotherapy resistance compared to ALDH-negative cells [14, 15]. Additionally, high ALDH activity at diagnosis predicts relapse in a subset of AML patients [16]. Among the 19 ALDH isoforms identified in humans, the most prominent are ALDH1 isoforms (ALDH1A1-3, ALDHB1, and ALDH1L1&2) and the ALDH2 isoform. ALDH1 family isoforms are located in the cytoplasm, whereas ALDH2 is located in the mitochondria [17, 18].

In this report, we investigated the effect of BM stromal cells on AML cells by determining the expression and activity of ALDH in AML cells. We also studied the signaling pathways activated and the therapeutic targets that contribute to chemotherapy resistance in AML cells. We identified specific inhibitors that could be used in combination with standard chemotherapy for the treatment of AML patients.

## Methods

### Cell culture of primary MSCs, leukemia cell lines, and patient samples

HL-60 cells were purchased from ATCC®, and MOLM-13 cells were obtained from The University of Texas MD Anderson Cancer Center Cell Line core facility. Both cell lines were cultured in Roswell Park Memorial Institute (RPMI; Media Tech, Inc., Manassas, VA) with 10% fetal bovine serum (FBS) (and 1% penicillin/streptomycin. OCI-AML3 cells were a kind gift from Dr. Mark Minden at Ontario Cancer Institute, Toronto, Canada. HEK293T cells were

purchased from ATCC® (Cat# CRL-3216) and cultured in DMEM high-glucose formulation (Media Tech) with 10% FBS and 1% penicillin/streptomycin.

Human MSCs were isolated from BM harvested from healthy donors for use in allogeneic stem cell transplantation. Donors were recruited from January 2014 through December 2019 at the Stem Cell Transplantation and Cellular Therapy Department at MD Anderson Cancer Center. All donors were healthy individuals between the ages of 25 and 45 and were recruited according to a protocol approved by the MD Anderson institutional review board. All study participants provided written informed consent as per the Declaration of Helsinki. Human MSCs were cultured in MSC Growth Medium 2 (Cat# C-28009, PromoCell®, Heidelberg, Germany) with 1% penicillin/streptomycin. Peripheral blood mononuclear cells (PBMCs) of patients obtained from the leukemia department according an institutional review board (IRB) approved protocol, between January 2017 and December 2019. PBMCs were cultured in RPMI containing 10% FBS and 1% penicillin/streptomycin. Tests for *Mycoplasma* contamination of MSCs and HEK293T cells are performed in our laboratory every 4–6 months.

## Flow cytometry assay

Flow cytometry analysis of OCI-AML3 cells cultured alone or co-cultured with MSCs was performed. MSCs isolated from normal donor–derived and AML patient–derived BM specimens were subjected to trypsin and washed once with phosphate-buffered saline (PBS). The cells were then incubated for 20 minutes with 10 μL of fluorochrome-conjugated antibodies. The antibody conjugates used were anti-CD45 conjugated with APC (Cat# 304038, BioLegend®, San Diego, CA) and anti-CD90 antibody conjugated with APC/Alexafluor 750 (Cat# B36121, Becton Dickinson Biosciences, Franklin Lakes, NJ). 4′, 6-Diamino-2-phenylindole (DAPI) was used to exclude dead cells (Cat# D1306, ThermoFisher Scientific, Waltham, MA). After incubation, the cells were washed once with PBS containing 0.5 μg/mL DAPI and analyzed on an LSR-II flow cytometer (Becton Dickinson Biosciences). Twenty thousand events were acquired for each sample. All flow cytometry data were analyzed by FlowJo software (FlowJo, LLC, Ashland, OR).

## ALDH activity assay

ALDH assay reagent was prepared according to the manual of the ALDEFLUOR™ Kit (Cat #01700, STEMCELL™ Technologies, Vancouver, Canada) and used to measure ALDH activity in AML cells. The assay was performed as per the manufacturer's recommendations. Briefly, fresh AML cell samples were prepared according to standard procedures. The AML cell concentration was adjusted to 1 million cells/mL of ALDEFLUOR™ buffer. One test tube and one control tube were used for each sample. Five microliters of the activated ALDE-FLUOR™ reagent per milliliter of sample was added to each sample test tube. The cells were mixed well, and 0.5 mL of the cell mixture was immediately transferred to the control tube containing DEAB (an ALDH inhibitor). The test tubes with cells were then incubated for 30 minutes at 37˚C in the dark. The cells were washed once with ALDEFLUOR™ buffer and stained with CD90 (to exclude MSCs during analysis) at 4˚C for 30 minutes. The final wash was performed using ALDEFLUOR™ buffer containing 1 μg/mL DAPI (to exclude dead cells). The cell pellets were re-suspended with 0.3 mL ALDEFLUOR™ buffer and analyzed by flow cytometry.

To determine the effect of ALDH2 inhibition on total ALDH activity, the cells were treated with the ALDH2 inhibitors diadzin (Cat# CS-4237, ChemScene, Monmouth Junction, NJ) and CVT-10216 (Cat # SML1366-5 mg, Sigma Aldrich, St. Louis, MO). ALDH activity was measured as described above.

## Protein analysis by Western blotting

Cells were lysed in RIPA buffer at $3\times10^5$/50 μL density. Protein concentrations were determined using the Bradford protein assay. Laemmli buffer was added to protein lysates at a 1:1 ratio. The lysates were loaded onto 4–15% Mini-PROTEAN® TGX™ Precast Protein Gels (Cat# 4561086, Bio-Rad, Hercules, CA), and proteins were subsequently transferred onto a polyvinylidene fluoride membrane. The membrane was blocked with 5% milk in 0.05% Tween-20 in PBS (PBS-T) to prevent nonspecific binding of antibodies. Primary antibody incubation was performed in PBS-T with 1% milk at 4˚C overnight (refer to S1 Table for list of primary antibodies used). IRDye® 680RD donkey anti-rabbit IgG or IRDye® 800CW goat anti-mouse IgG (LI-COR Biosciences®, Lincoln, NE) was incubated with the membranes for 1 hour at room temperature in PBS-T with 1% milk. The membranes were washed 3 times with PBS-T and scanned using an Odyssey Western blot scanner (LI-COR Biosciences®). All protein quantification was performed using LI-COR image analysis software.

## shRNA knockdown of TGF-β1 expression

Lentiviral-mediated short-hairpin RNA (shRNA) was used for stable knockdown of TGF-β1 in human BM-derived MSCs. shRNA lentiviral vectors (NM_000660, XM_011527242; Clone ID: TRCN0000003318; Sequencing Primer: 5'—AAACCCAGGGCTGCCTTGGAAAAG—3'; Vector Map: pLKO.1) were purchased from GE Healthcare Dharmacon, Inc. (Lafayette, CO). Lentiviral pLKO.1 Empty Vector (Cat# RHS4080, GE-Dharmacon) was used as control. HEK293T cells were transfected with each TGF-β1 shRNA construct along with packaging vectors pMD2.G (0.5 μg) and psPAX2 (1.5 μg) (Addgene, Inc., Watertown, MA) using Jet Prime Reagent (Polyplus-transfection®, Illkirch-Graffenstaden, France) according to the manufacturer's guidelines. The medium containing lentivirus was collected 72 hours after transfection and incubated with BM-MSC cells for 24 hours. The transduced cells were selected using puromycin (0.5 μg/mL) for 3 days, and TGF-β1 mRNA and protein knockdown efficacy was determined by quantitative polymerase chain reaction (qPCR) or Western blotting, respectively. The plasmid TRCN0000003318 (RHS4533-EG7040, GE-Dharmacon) provided the best knockdown efficacy.

## Total RNA isolation and gene expression by real-time PCR

Five million cells were lysed in Trizol reagent overnight at -80˚C, total RNA was isolated by ethanol precipitation, and real-time PCR (RT-PCR) was performed with a QuantStudio3 (Applied Biosystems®, Foster City, CA) instrument using TaqMan Fast Universal PCR Master Mix (Applied Biosystems®) as described before [19]. All samples were run in triplicate. The relative fold increase of specific RNA was calculated by the comparative cycle of threshold detection method, and values were normalized to glyceraldehyde 3-phosphate dehydrogenase (GAPDH). Fold changes in gene expression were calculated using the 2-ddC method. All primer pairs for human samples were purchased from ThermoFisher Scientific (S2 Table).

## AML cell separation by fluorescence-activated cell sorting

OCI-AML3 and BM-MSC cells were seeded at a 5:1 ratio in 75 cm$^2$ flasks containing RPMI medium with 10% FBS and cultured for 3–5 days. On the day of cell separation, the non-adherent fraction of OCI-AML3 cells was separated by removing the supernatant. The adherent fraction was washed once with PBS to remove traces of FBS, which could inhibit trypsin in the next steps. The adherent AML and MSC fraction was then detached by trypsinization and mixed with the previously collected non-adherent cells. After a single wash with PBS, the cells

were stained with APC-Cy7-conjugated anti-CD90 (Cat # 16699531, Becton Dickinson Biosciences) and fluorescein isothiocyanate-conjugated anti-CD45 (Cat # 304038, BioLegend®) antibodies. After a single wash with PBS containing DAPI solution, the cells were subjected to fluorescence-activated cell sorting (FACS) using the BD FACS Aria-II cell sorter (Becton Dickinson Biosciences). To isolate AML cells, the cells were gated on a CD45+CD90- fraction using FACSDiva software. The sorted cells were used for further analysis including gene expression studies.

## Gene expression analysis by RNA sequencing

Samples were sequenced on the HiSeq Sequencing System by the Sequencing and Microarray Core Facility at MD Anderson. Sequence reads were mapped to human genomics (build hg19) with bowtie2 aligner using RSEM software. The EdgeR package in R software was used to compare the differential gene expressions in OCI-AML3 cells cultured alone versus those cultured with BM-MSCs. Genes with adjusted p values less than 0.05 and absolute fold changes larger than 2 were considered significant. Changes in gene expression in pathways of interest were visualized using Ingenuity® Pathway Analysis (IPA, Ingenuity Systems, Inc. Redwood City, CA). To investigate the relationship between the significant genes differentially expressed in the presence of BM-MSCs, we used AML expression data obtained from the TCGA dataset. A scatterplot with log2 fold change from expression in co-culture and TCGA correlation coefficients was plotted using the cBioPortal data analysis tool. The array data have been deposited in the Gene Expression Omnibus (GEO), identified by the accession number GSE152996.

## p38 MAPK inhibition

The p38 MAPK inhibitor SB-203580 was purchased from R&D Systems (Cat# 1202, Minneapolis, MN). OCI-AML3 cells were cultured with or without BM-MSCs and treated with p38 MAPK inhibitor (SB-203580; 2 μM) for 3 days; similarly, OCI-AML3 cells were treated with p38 MAPK inhibitor in the presence or absence of recombinant TGF-β1 (5 ng/mL). ALDH activity was measured using ALDEFLUOR®, as described above. To determine the effect of p38 inhibition on ALDH2 expression, OCI-AML3 cells cultured with or without BM-MSCs were treated with SB-203580 at 2 μM for 3 days, and ALDH2 expression was measured by Western blotting.

## ALDH2 inhibition in combination with standard chemotherapy

To assess the combined effect of ALDH2 inhibition and chemotherapy on AML cells, OCI-AML3 cells were cultured at 0.25 million cells/mL density with or without BM-MSCs (100 000 cells) and treated with cytarabine (Ara-C; 2.5 μM; obtained from the MD Anderson pharmacy), alone or in combination with CVT-10216 (5 μM), for 48 hours. Cells were stained with Annexin V and analyzed on an LSR-II flow cytometer.

## Leukemia engraftment and growth rate

To investigate the effect of stromal cells on leukemia engraftment and growth, we implanted 1 million Molm13 cells expressing firefly luciferase and green fluorescent protein subcutaneously, with or without 1 million BM-MSCs and 100 μL Matrigel, into Nonobese Diabetic/ Severe Combined Immunodeficiency (NOD/SCID) mice. Leukemia engraftment and growth rate assessment was performed at 1 and 2 weeks as previously described [20].

## Animal study approval and mice handling

All experiments performed with mice were in compliance with the standards of care of the MD Anderson Institutional Animal Care and Use Committee (IACUC) as well as in accordance with IACUC-approved protocols. Briefly, the NOD/SCID mice were purchased from The Jackson Laboratory (Bar Harbor, ME). The mice were maintained in designated cages at the animal housing facility provided by the Department of Veterinary Medicine and Surgery at MD Anderson. The mice were fed and sheltered according to the Association for Assessment and Accreditation of Laboratory Animal Care accreditation standards. Mice were euthanized in carbon dioxide chambers when the average tumor size became greater than 2 cm in diameter, which was considered as the endpoint of the study. No mice died before reaching the desired experimental endpoints.

## Statistical analyses

For survival analysis, we used the Kaplan-Meier estimator to estimate the survival function and the log-rank test to evaluate the statistical significance. To compare the difference between two independent groups, we used the Mann-Whitney U test or Student's $t$ test to examine the statistical significance. For the comparison of two groups with paired data, the paired Student's $t$ test was used. We performed one-way analysis of variance (ANOVA) with Tukey's HSD post-hoc test to test the significance in comparisons with more than 2 groups. A linear model with interaction term was also used to evaluate the significance with more than two factors in the experiment. P value less than 0.05 was regarded as statistically significant.

## Ethics approval and consent to participate

This study was approved by an Institutional Review Board at MD Anderson. All samples were obtained with informed written consent from adult participants in accordance with the Declaration of Helsinki. No minors were included in our study. All animal experiments were in compliance with a protocol approved by the MD Anderson Cancer Center Institutional Animal Care and Use Committee.

# Results

## BM-MSCs induce leukemia growth in vivo by inducing ALDH activity in AML cells

BM stroma contributes to AML progression and chemotherapy resistance [3, 20, 21]. To investigate the effect of stromal cells on AML growth, we implanted Molm13 cells, with or without BM-MSCs, subcutaneously into NOD/SCID mice. Bioluminescence imaging revealed that Molm13 cells implanted together with BM-MSCs grew 8-fold faster than Molm13 cells implanted alone, suggesting that BM-MSCs support AML cell growth (Fig 1A and 1B).

To investigate the effect of BM stromal cells on AML cells, we co-cultured OCI-AML3 and HL60 AML cells with or without BM-MSCs for 3 or 5 days and measured ALDH activity. The percentage of ALDH+ cells increased from 31% ± 6% to 94% ± 0.5% when OCI-AML3 cells were co-cultured with BM-MSCs compared to being cultured alone. In HL60 cells, co-culturing with BM-MSCs increased the percentage of ALDH+ cells from 22% ± 1% to 67% ± 1% (Fig 1C–1H).

To validate stroma-induced ALDH activity in primary AML cells, we analyzed ALDH activity in peripheral blood and BM samples derived from AML patients and found that the percentage of AML cells varied between patients and did not correlate with age, sex, white blood cell count, or blast percentage (S3 Table). Next, patient-derived primary AML cells were

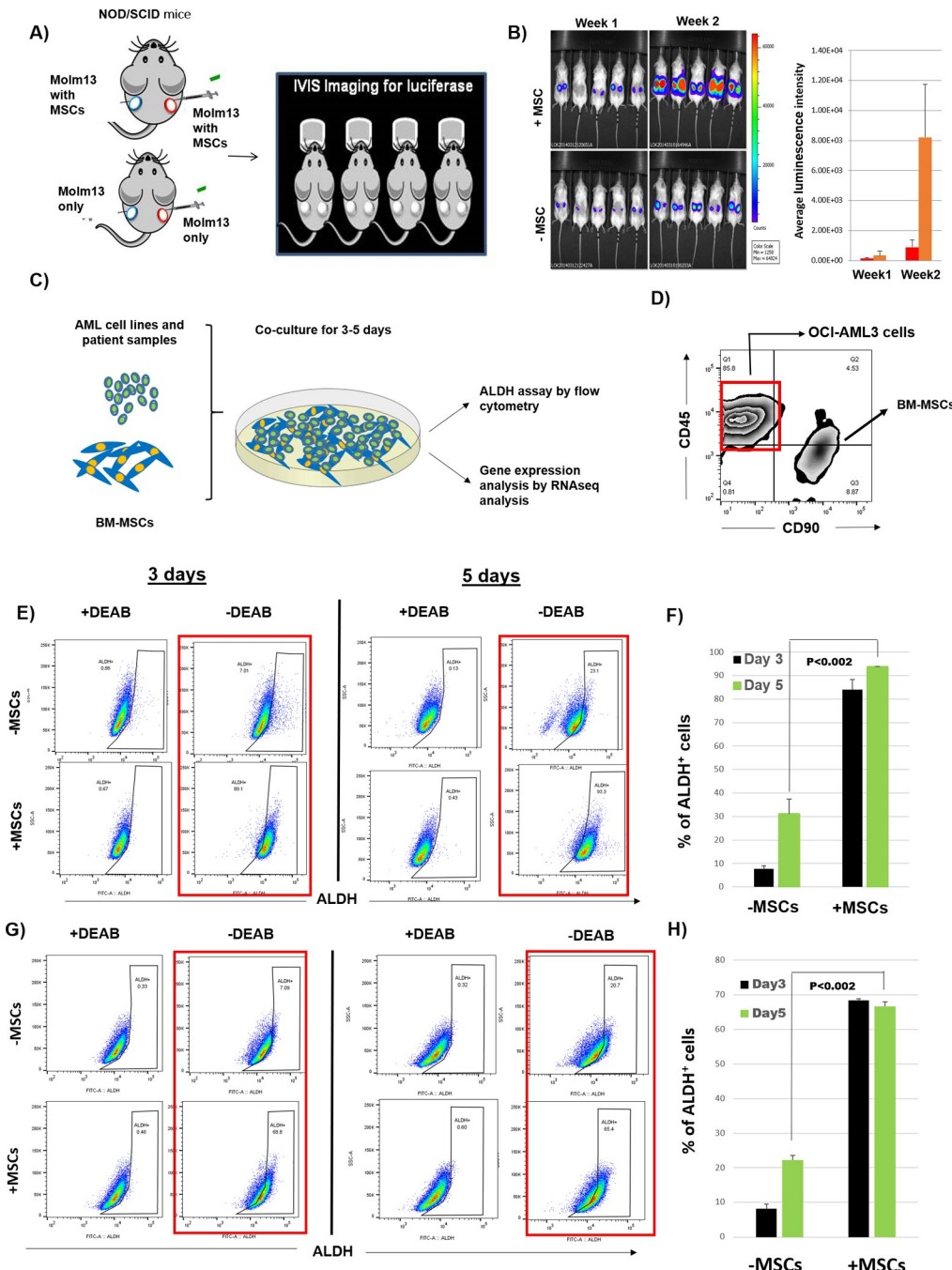

**Fig 1. BM-MSCs induce ALDH activity in AML cells and enhance engraftment in mice. (A, B)** One million firefly luciferase-expressing AML cells (Molm13) were implanted subcutaneously, with or without 1 million BM-MSCs and 100 μL of Matrigel, into NOD/SCID mice. Bioluminescence imaging was performed at 1 and 2 weeks to check leukemia engraftment and growth rate. **(C)** AML cell lines and patient samples were cultured with or without BM-MSCs for 3–5 days. ALDH activity in AML cells was measured using ALDEFLUOR® assay. Gene expression analysis was performed by RNA sequencing. **(D)** Fluorescence-activated cell sorting of OCI-AML3 cells was performed based on phenotype (CD45$^+$, CD90$^-$) to distinguish them from BM-MSCs (CD90$^+$, CD45$^-$). **(E)** OCI-AML3 cells were cultured with or without BM-MSCs for 3 or 5 days. Cells were stained with ALDEFLUOR®, CD45, and CD90. During FACS analysis, MSCs (CD90$^+$, CD45$^-$) were gated and ALDH activity was measured in OCI-AML3 cells by flow cytometry. Data were analyzed on FlowJo software. **(F)** Histogram representation of the percentage of ALDH$^+$ OCI-AML3 cells corresponding to the experiment done in **E**. **(G)** The same experiment was done as in **E**, using HL60 AML cells. **(H)** Histogram representation showing the percentage of ALDH$^+$ HL60 cells corresponding to the experiment done in **G**. Data are plotted as the mean value with error bars representing standard error. For **B**, **F**, and **H** a linear model with interaction term was used to evaluate the significance. N = 3 for each group.

cultured with or without MSCs for 3 days and ALDH activity was measured in the AML cells. We found that co-culture with MSCs significantly induced ALDH activity in AML cells in all 8 patient samples (Fig 2A and 2B). This indicates that BM-MSCs support AML cell growth in vivo and induce the ALDH+ stem cell phenotype in AML cells.

## ALDH isoforms are differentially expressed in AML cells in the presence of stromal cells

To identify the ALDH isoforms responsible for increasing ALDH activity in AML cells in the presence of stromal cells, we performed gene expression analysis of the 19 ALDH isoforms by real-time RT-PCR (primers listed in S2 Table). We found differential expression of ALDH isoforms in AML cells cultured with or without BM-MSCs. Specifically, ALDH1L2 and ALDH2 expressions were upregulated 3- to 5-fold in OCI-AML3 cells co-cultured with BM-MSCs compared to OCI-AML3 cells cultured alone (Fig 2C). To determine the prognostic significance of these isoforms, we analyzed ALDH1L2 and ALDH2 expression in the TCGA AML dataset, which revealed that ALDH1L2 and ALDH2 are upregulated in 8% and 12% of AML cases, respectively (S1 Fig). However, increased expression of ALDH2, but not ALDH1L2, confers a worse prognosis and lower survival rate, suggesting that ALDH2 is a key factor promoting AML disease progression (Fig 2D).

## TGF-β1-associated gene signature is activated in OCI-AML3 cells co-cultured with BM-MSCs

To investigate the mechanism of stroma-induced ALDH activity in AML cells, we co-cultured OCI-AML3 cells with or without BM-MSCs for 3 days. OCI-AML3 cells were FACS sorted and gene expression analysis was performed by RNA sequencing. Analysis of differentially expressed genes by the Ingenuity® pathway analysis tool revealed activation of a TGF-β1- associated gene signature in OCI-AML3 cells co-cultured with BM-MSCs compared to OCI-AML3 cells cultured alone (Fig 3A and 3B). To validate this, we performed RT-PCR for genes that are differentially regulated by TGF-β1. Genes that are positively regulated by TGF-β1 were upregulated and genes that are negatively regulated by TGF-β1 were downregulated in OCI-AML3 cells co-cultured with MSCs compared to cells cultured alone (Fig 3C). Hence, TGF-β1-regulated transcriptional activity is upregulated in AML cells co-cultured with BM-MSCs.

To determine whether TGF-β1 is derived from stromal cells and regulates ALDH activity in AML cells, we knocked down TGF-β1 in BM-MSCs. We found that 2 of 5 shRNA sequences showed highest TGF-β1 knockdown efficacy in BM-MSCs (S2 Fig). We then co-cultured OCI-AML3 cells with control (scrambled shRNA) or TGF-β1-knockdown BM-MSCs for 3 days and measured ALDH activity in OCI-AML3 cells. Remarkably, the percentage of ALDH+ cells decreased by ~4-fold in OCI-AML3 cells co-cultured with TGF-β1-knockdown BM-MSCs compared to OCI-AML3 cells cultured with control BM-MSCs (Fig 3D). Therefore, knockdown of TGF-β1 inhibits stroma-induced ALDH activity, further solidifying the hypothesis that TGF-β1 secreted by stromal cells is directly involved in the increase in ALDH activity in AML cells.

## Recombinant TGF-β1 induces ALDH activity in AML cells

It has been well established that TGF-β1 signaling is involved in AML-BME interactions [5, 22]. TGF-β is highly expressed in BM-MSCs and its expression is further enhanced in co-culture with leukemia cells [22]. TGF-β has been shown to induce quiescence and a stem-like phenotype in solid tumors as well as in leukemia [8]. To test whether TGF-β1 induces ALDH

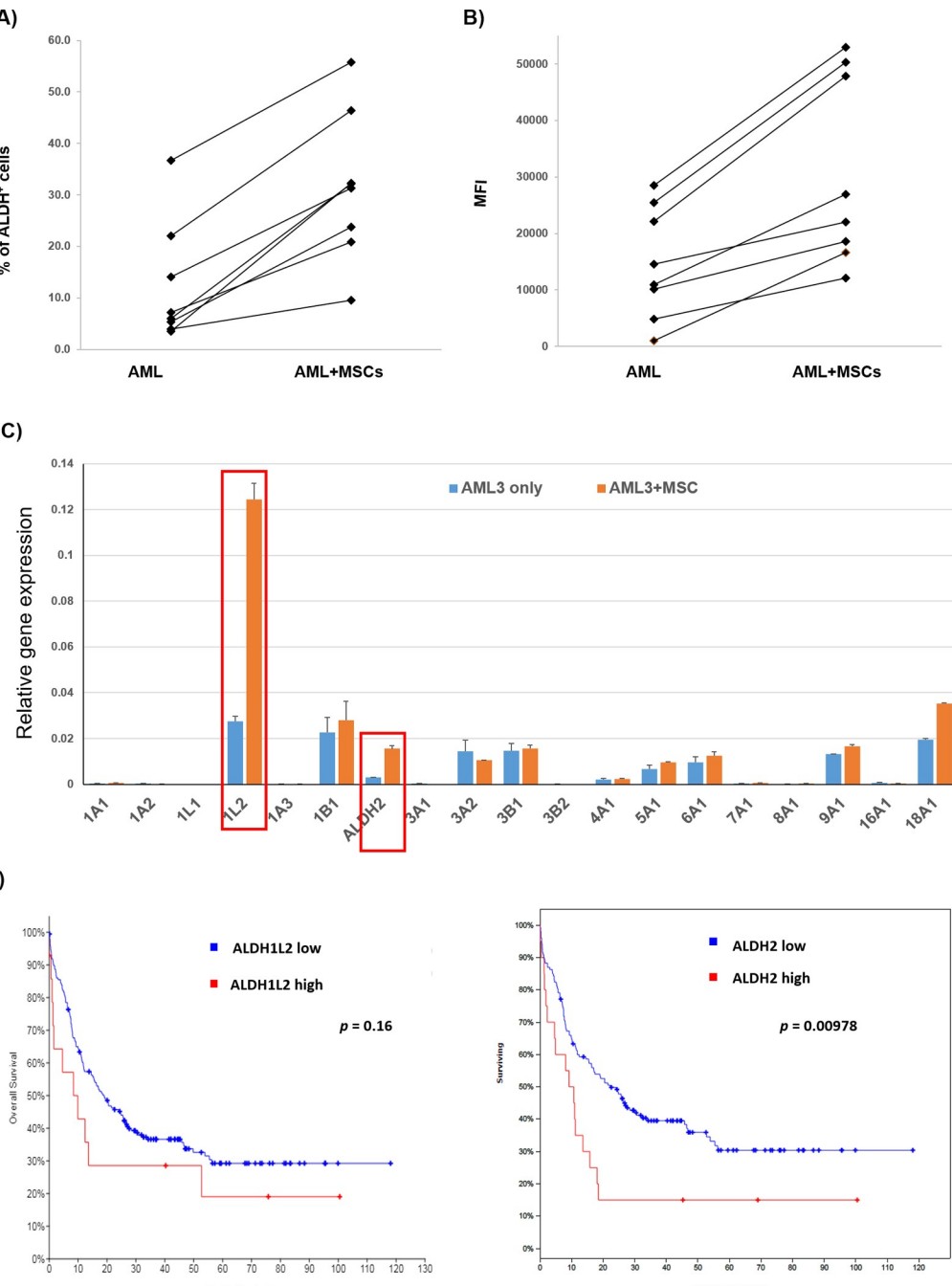

**Fig 2. Differential expression of ALDH isoforms in AML cells co-cultured with BM-MSCs. (A, B)** Patient-derived primary AML cells from bone marrow and peripheral blood samples were cultured with or without MSCs for 3 days and ALDH activity was measured using ALDEFLUOR® assay by flow cytometry. Data show a graphical representation of ALDH mean fluorescence intensity (MFI) and percentage of ALDH⁺ AML patient samples when co-cultured with stromal cells. **(C)** Total RNA was extracted from 5 million AML cells. RT-PCR was performed to analyze gene expression of different ALDH isoforms using primers listed in S2 Table. All samples were run in triplicate. The relative fold increase of specific RNA was calculated by the comparative cycle of threshold detection method, and values were normalized to GAPDH. Fold changes in gene expression were calculated using the 2-ddC method after normalization to GAPDH. Data are plotted as mean values with error bars representing standard error. **(D)** AML expression data for ALDH isoforms and Kaplan-Meier survival analysis were obtained from the TCGA dataset. For **A** and **B**, the paired *t* test was used. For **C**, the Mann-Whitney U test/Student's *t* test was used (N = 3). For **D**, the log-rank test was used to test the significance.

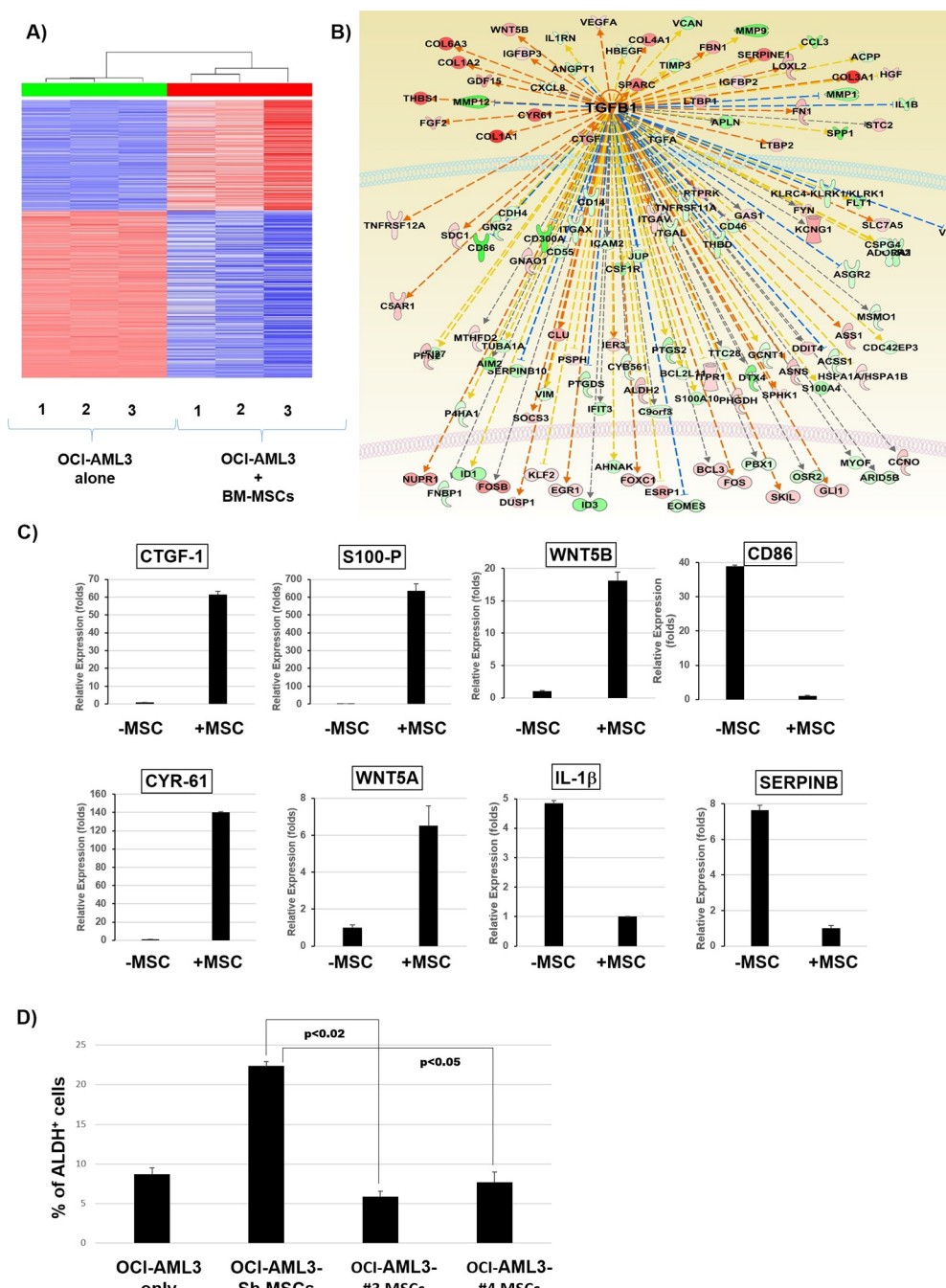

**Fig 3. Activation of TGF-β1 gene signature in OCI-AML3 cells co-cultured with BM-MSCs. (A, B)** OCI-AML3 cells were cultured with or without BM-MSC cells for 3 days. OCI-AML3 cells were FACS sorted to separate them from BM-MSCs and the gene expression analysis was performed by RNA sequencing. Samples were sequenced on the HiSeq Sequencing System. Sequence reads were mapped to human genomics (build hg19) with bowtie2 aligner using RSEM software. R software was used to compare the differential expression between MSC co-culture samples and OCI-AML3 controls. Genes with adjusted p values less than 0.05 and absolute fold changes larger than 2 were considered significant. Analysis of differentially expressed genes using the Ingenuity® pathway analysis tool revealed activation of a TGF-β1-associated gene signature. **(C)** Real-time PCR analysis was performed to analyze the expressions of the indicated genes that are differentially regulated by TGF-β1 in OCI-AML3 cells co-cultured with BM-MSCs compared to OCI-AML3 controls. **(D)** TGF-β1 knockdown MSCs were generated by transfection with plasmid-containing lentiviruses and co-cultured with OCI-AML3 cells for 3 days. ALDH activity was measured by flow cytometry in OCI-AML3 cells cultured with TGF-β1 knockdown MSCs compared to OCI-AML3 cells cultured alone or with control BM-MSCs. Data are plotted as mean values with error bars representing standard error. For **C**, the Mann-Whitney U test or Student's *t* test was used. For **D**, one-way ANOVA with Tukey's HSD post-hoc test was used.

activity in AML cells leading to a stem-like phenotype, we treated OCI-AML3 and HL60 cells with or without recombinant TGF-β1 (5 ng/mL) for 3 days and measured ALDH activity. Interestingly, treatment with recombinant TGF-β1 increased the percentage of ALDH$^+$ cells from 30% ± 5% to 90% ± 3% and from 10% ± 4% to 25% ± 3% in OCI-AML3 and HL60 cells, respectively, suggesting that TGF-β1 directly regulates ALDH activity in AML cells (Fig 4A and 4B).

To validate that TGF-β1 regulates downstream transcriptional activity in AML cells, we measured mRNA expression of TGF-β1 target genes in OCI-AML3 cells treated with recombinant TGF-β1. We found that TGF-β1 target genes were upregulated in cells treated with recombinant TGF-β1 compared to untreated controls (Fig 4C). Interestingly, we also found that ALDH2, which was upregulated in AML cells upon co-culture with BM-MSCs, was also upregulated at the mRNA and protein levels in cells treated with recombinant TGF-β1 (Fig 4C and 4D).

## TGF-β non-canonical pathway is involved in stroma-induced ALDH activity in AML cells

TGF-β has been reported to alter downstream target gene expression through either its canonical pathway or its non-canonical pathway involving p38 and ERK [6, 23]. To delineate the specific signaling pathway involved in TGF-β1-induced ALDH activity in AML cells, we treated OCI-AML3 cells with or without cell culture supernatants from BM-MSCs for 0, 30, 60, and 120 minutes and measured phosphorylation of transcription factors that mediate TGF-β1 canonical and non-canonical pathways by Western blotting. Interestingly, we couldn't detect any phosphorylation of Smad2 or Smad3 transcription factors in OCI-AML3 cells treated with supernatants derived from BM-MSCs. To confirm that BM-MSCs do not induce the canonical TGF-β pathway in AML cells, we measured phosphorylation of Smad2 and Smad3 in OCI-AML3 cells treated with recombinant TGF-β1 and still could not find any activity for these 2 proteins (Fig 4E). Next, we tested phosphorylation of p38, which is activated by TGF-β through its non-canonical pathway. Interestingly, we found strong activity for phospho-p38 in OCI-AML3 cells treated with supernatants from BM-MSCs compared to cells treated with medium alone. We also found increased phosphorylation of ERK in OCI-AML3 cells treated with BM-MSC supernatants, suggesting activation of the Raf-MEK-ERK pathway in these cells. We validated this by treating OCI-AML3 cells with recombinant TGF-β1 and found a time-dependent increase in p38 phosphorylation (Fig 4E).

p38 MAPK is a downstream target of the TGF-β non-canonical/non-Smad signaling pathway, which phosphorylates p38; p38 then activates other downstream signals leading to regulation of gene expression [22, 24, 25]. To investigate the role of phospho-p38 in TGF-β1-mediated, stroma-induced ALDH2 expression in AML cells, OCI-AML3 cells were treated with or without the p38 MAPK inhibitor SB-203580 (5 μM) and ALDH2 expression was measured by Western blotting. We found that inhibition of p38 dramatically inhibited ALDH2 expression in OCI-AML3 cells, even in the presence of BM-MSC supernatant or recombinant TGF-β1 (Fig 4F). This indicates that stromal cells induce ALDH2 expression in AML cells through the non-canonical TGF-β/p38 MAPK pathway.

## p38 MAPK inhibition significantly inhibits ALDH activity in the presence of TGF-β or MSCs

To investigate the effect of p38 inhibition on ALDH2 activity, we treated OCI-AML3 cells with p38 MAPK inhibitor (5 μM) in the presence or absence of recombinant TGF-β1 (5 ng/mL) and measured ALDH activity. We found that p38 MAPK inhibition decreased the percentage

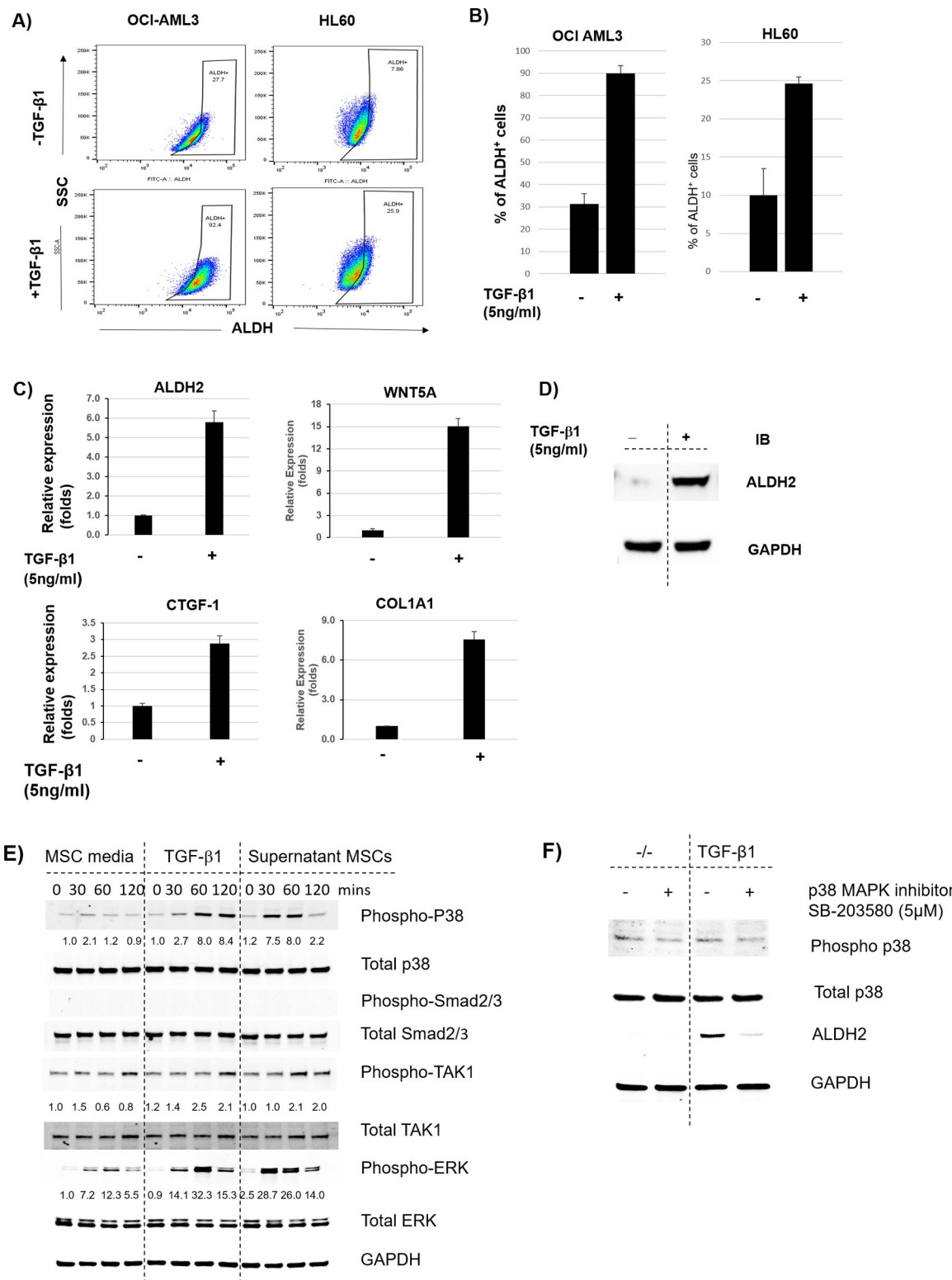

**Fig 4. TGF-β1 non-canonical pathway induces expression of ALDH2 and other downstream targets in AML cells. (A)** OCI-AML3 and HL60 cells were treated with recombinant TGF-β1 (5 ng/mL) for 3 days. Cells were stained with ALDEFLUOR® and ALDH activity was measured. **(B)** Histogram representation of the percentages of ALDH⁺ cells in OCI-AML3 and HL60 AML cells treated with recombinant TGF-β1 in **A. (C)** OCI-AML3 cells were treated with recombinant TGF-β1 (5 ng/mL). RT-PCR was performed to analyze

mRNA expression of ALDH and the downstream TGF-β targets WNT5A, CTGF1, and COL1A1. Relative fold increase values in gene expression were normalized to GAPDH. Data are plotted as mean values with error bars representing standard error. (**D**) OCI-AML3 cells were treated with recombinant TGF-β1 (5 ng/mL) for 3 days. Protein lysates were harvested and Western blotting was performed to analyze ALDH2 expression in treated cells compared to untreated controls. (**E**) OCI-AML3 cells were cultured in RPMI overnight. MSC cells were centrifuged and the MSC supernatant and TGF-β1 (5 ng/mL) were added to stimulate OCI-AML3 cells. Western blot analysis was performed at the indicated time points to analyze protein expression of downstream targets of the TGF-β canonical and non-canonical signaling pathways. (**F**) OCI-AML3 cells were treated with the p38 MAPK inhibitor SB-203580 (5 μM) in the presence or absence of recombinant TGF-β1 (5 ng/mL). Protein lysates were harvested and p38 and ALDH2 expression was analyzed by Western blotting. Due to the similarity in molecular weight of several proteins tested, samples were run on different gels. Original raw western blot images are available as S1 Raw images. All membranes were scanned using Odyssey Western blot scanner (LI-COR Biosciences®) and images were obtained using the corresponding software. For **B** and **C**, the Mann-Whitney U test and Student's $t$ test were used, respectively (N = 3).

of ALDH$^+$ OCI-AML3 cells from 43% ± 5% to 20% ± 1%. Moreover, in the presence of recombinant TGF-β1, p38 MAPK inhibitor decreased the percentage of ALDH$^+$ OCI-AML3 cells from 96% ± 1% to 22% ± 1% (Fig 5A). Similarly, the addition of p38 MAPK inhibitor decreased the percentage of ALDH$^+$ OCI-AML3 cells co-cultured with MSCs (Fig 5B). These results solidify the notion that the non-canonical TGF-β pathway regulates ALDH activity in AML cells in the BME and that inhibition of p38 MAPK decreases TGF-β- or stroma-induced ALDH2 activity in AML cells.

## ALDH2 inhibitors significantly decrease MSC-induced ALDH activity in AML cells

ALDH2 overexpression has been associated with several malignancies and diseases, leading to the development of specific inhibitors with potential therapeutic benefits [18, 26, 27]. To validate

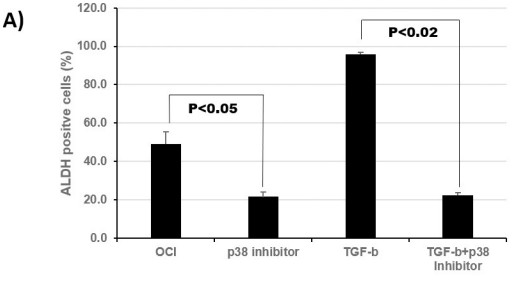
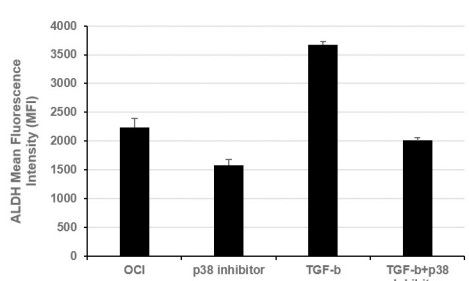

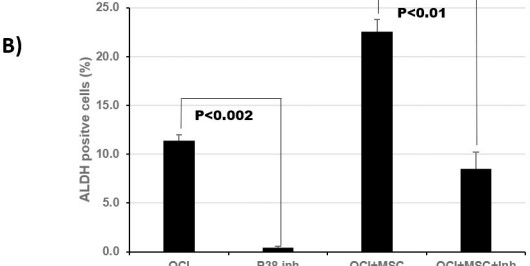
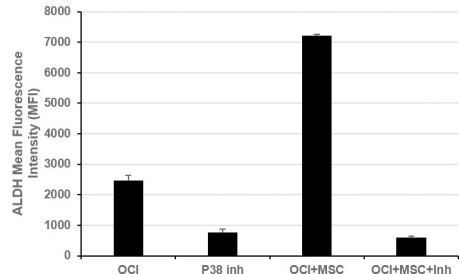

**Fig 5. p38 MAPK inhibitor decreases ALDH2 expression in AML cells in the presence of TGF-β1 or stromal cells. (A)**. OCI-AML3 cells were treated with the p38 MAPK inhibitor SB-23580 (5 μM) in the presence or absence of recombinant TGF-β1 (5 ng/mL). ALDH activity was measured by flow cytometry using ALDEFLUOR® assay in comparison to untreated controls. (**B**) OCI-AML3 cells were cultured with or without BM-MSCs and treated with p38 MAPK as in **A.** ALDH activity was measured in treated cells in comparison with untreated controls. Data are plotted as mean values with error bars representing standard error via one-way ANOVA with Tukey's HSD post-hoc test (N = 3).

that stroma-induced ALDH activity is mostly due to the ALDH2 isoform, we cultured OCI-AML3 cells and treated them with the ALDH2 inhibitors diadzin or CVT-10216. Strikingly, treatment with ALDH2-specific inhibitors significantly inhibited ALDH activity in OCI-AML3 cells dose-dependently. The percentage of ALDH[+] cells decreased from 29% ± 1% in untreated cells to 4% ± 0.5% in cells treated with 50 μM of diadzin. Similarly, the percentage of ALDH[+] cells dropped from 16% ± 5% in untreated cells to 4% ± 0.2% in cells treated with 2 μM of CVT-10216 (Fig 6A and 6B). Next, OCI-AML3 cells were treated with diadzin (5 μM) or CVT-10216 (1 μM) in the presence or absence of recombinant TGF-β1 (5 ng/mL) for 3 days. As expected, when treated with recombinant TGF-β1, the percentage of ALDH[+] cells in OCI-AML3 cells increased from 23% ± 3% to 63% ± 2%. However, ALDH[+] cells decreased from 63% ± 2% to 45% ± 2% when treated with diadzin in addition to recombinant TGF-β1, suggesting that diadzin inhibits ALDH activity even in the presence of TGF-β1. Similarly, we found an ~50% reduction in ALDH activity in OCI-AML3 cells treated with CVT-10216, even in the presence of recombinant TGF-β1 or BM-MSCs (Fig 6A and 6B). Dotplots and mean fluorescent intensity corresponding to these experiments are shown in S3 Fig. This indicates that stroma-mediated, TGF-β1-induced ALDH activity in AML cells can be decreased by ALDH2 inhibitors.

## ALDH2 inhibitors sensitize AML cells to standard chemotherapy

To evaluate the therapeutic significance of ALDH2 inhibition in AML, we tested the effect of CVT-10216 in combination with standard chemotherapy on AML cell death. We cultured OCI-AML3 cells with or without BM-MSCs and treated them with cytarabine (2.5 μM) alone or in combination with CVT-10216 (2 μM) for 48 hours. The cells were stained with Annexin V and DAPI and analyzed by flow cytometry to measure treatment-induced apoptosis and cell death. Interestingly, treatment with CVT-10216 significantly improved cytarabine-induced cell death in AML cells. The combination of CVT-10216 and cytarabine in the presence of BM-MSCs induced cell death in 39% ± 6% of OCI-AML3 cells compared to 24% ± 5% of cells treated with cytarabine alone (Fig 6C). The majority of the OCI-AML3 cells stained positive for Annexin V and DAPI, indicating that the mechanism of treatment-induced cell death is late apoptosis as opposed to early apoptosis or necrosis. Our results suggest ALDH2 is a potential therapeutic target in AML and that CVT-10216 sensitizes AML cells to conventional chemotherapy.

Collectively, we found that MSCs in the BME secrete TGF-β1 and exert its effects through a non-canonical/p38-dependent signaling pathway, leading to ALDH2 overexpression in AML cells. Consequently, AML cells acquire a stem-like phenotype, which promotes tumorigenicity and chemotherapy resistance (Fig 6D).

## Discussion

In the present study, we demonstrate that BM-MSCs contribute to AML progression and chemo-resistance by inducing an ALDH[+] stem-like phenotype in AML cells. Importantly, in-depth gene expression analysis of all 19 ALDH isoforms identified ALDH2 as the isoform most differentially expressed and primarily responsible for the stroma-induced increased ALDH activity, identifying it as a major contributor to the AML stem-like phenotype and observed chemotherapy resistance. These findings also underscore the importance of drug testing in leukemia cells in AML-MSC co-culture systems, which has been long advocated and practiced by our group [28–30].

Recent studies have identified TGF-β as an important mediator of AML-BME interactions and chemotherapy resistance [8, 31]. TGF-β expression is upregulated in stromal cells in the AML-BME and contributes to tumor growth and treatment resistance. Silencing of TGF-β in

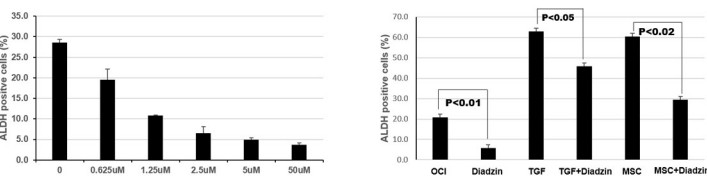

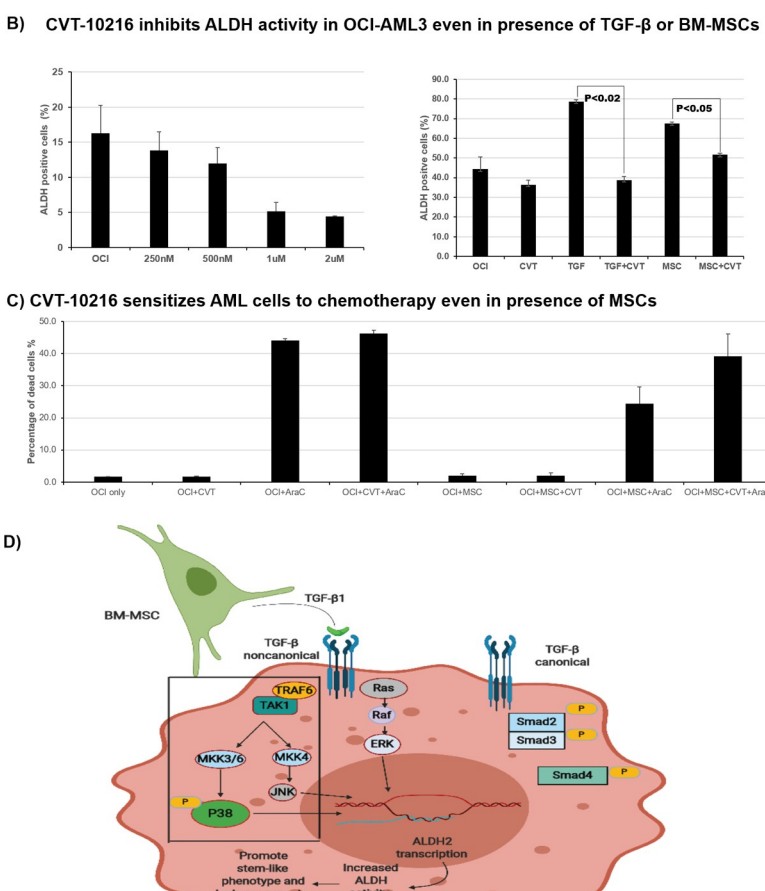

**Fig 6. ALDH2 inhibitors decrease stroma-induced ALDH activity in AML cells and sensitize them to chemotherapy. (A)** OCI-AML3 cells were cultured and treated with increasing concentrations of the ALDH2 inhibitor diadzin. ALDH activity was measured by ALDEFLUOR® assay using flow cytometry. OCI-AML3 cells were co-cultured with MSCs (200 000 cells/well density) and treated with diadzin (5 μM) for 3 days in the presence or absence of recombinant TGF-β1 (5 ng/mL). ALDH activity was measured as described above. **(B)** The same experiments were performed as in **A**, using CVT-10216 (at increasing concentrations and 1 μM) instead of diadzin. **(C)** OCI-AML3 cells were cultured alone or with MSCs (100 000 cells/well density) and treated with cytarabine (2.5 μM) alone or in combination with CVT-10216 (2 μM) for 48 hours. Cell samples were stained with Annexin V and analyzed by flow cytometry to measure treatment-induced cell death. Data are plotted as mean values with error bars representing standard error. **(D)** Simplified schematic representation of the interaction between stromal cells and AML cells leading to ALDH2 expression. BM-MSCs secrete TGF- β1, which induces ALDH2 expression in AML cells through the non-canonical/p38-dependent pathway, thereby promoting leukemogenesis. For **A**, **B**, and **C** one-way ANOVA with Tukey's HSD post-hoc test was used (N = 3).

stromal cells co-cultured with FLT3-ITD-AML enhances treatment-induced AML apoptosis and sensitizes AML cells to combination chemotherapy [32]. Moreover, TGF-β inhibition by small-molecule inhibitors or neutralizing antibodies is a strategy that has been explored to

counteract chemotherapy resistance in several malignancies, including AML [5]. Despite the aforementioned advances in elucidating the role of TGF-β in the progression of AML, the exact mechanism of TGF-β-mediated tumor progression and chemotherapy resistance remains unclear.

Our results confirm published reports of upregulation of the TGF-β1 gene signature in AML cells in the presence of stromal cells [5, 33]. We also show that downstream targets of TGF-β signaling, including ALDH2, are differentially expressed in AML cells in the presence of MSCs. Moreover, exposure of AML cells to recombinant TGF-β1 induced ALDH activity through ALDH2 expression, and knockdown of TGF-β1 in BM-MSCs led to a decrease in ALDH activity, indicating that TGF-β1 secreted by BM-MSCs induces a stem-like phenotype in AML cells. Additionally, through protein expression analysis, we delineated the specific signaling pathway involved in the aforementioned TGF-β1-mediated AML-BME interaction. We showed that TGF-β1 exerts its effect through a non-canonical/p38-dependent signaling pathway. Thus, our work uncovers a clear link between TGF-β1 secreted by MSCs and the acquisition of a stem-like phenotype of AML cells through ALDH2 overexpression; it also establishes the specific signaling mechanism involved in this interaction.

Aldehyde dehydrogenases constitute a group of enzymes that neutralize toxic aldehydes by converting them to carboxylic acids. To date, 19 different ALDH isoforms have been identified within the human genome. ALDH is a well-known marker for cancer stem cells and can be used to identify hematopoietic stem cells as well as leukemia stem cells [15]. ALDH plays a prominent role in several diseases and malignancies, and its expression is associated with a worse prognosis [18, 34]. Therefore, specific ALDH inhibitors with potential anti-tumor effects have been developed [17, 26, 35–37]. Aldehyde dehydrogenases have been identified as a therapeutic target to eliminate cancer stem cells and reduce tumor growth synergistically with standard chemotherapy in gynecologic malignancies [38]. In AML specifically, recent studies emphasize the role of ALDH in promoting tumor survival. One report demonstrates that the ALDH3A2 isoform is crucial for AML survival, as it protects cells from oxidative damage and consequent cell death. Additionally, ALDH3A2 inhibition is synthetically lethal when combined with inhibition of glutathione peroxidase-4, a known inducer of ferroptosis [39]. Although some effort has been made to identify the role of specific aldehyde dehydrogenases in promoting AML growth and therapy resistance, the molecular mechanism underlying this finding remains unexplored.

In this study, we examined the role of ALDH2 inhibitors in counteracting the stem-like phenotype and chemotherapy resistance acquired by AML cells in the presence of stromal cells. We found that the specific ALDH2 inhibitors diadzin and CVT-10216 significantly inhibit ALDH activity in a dose-dependent manner and sensitize AML cells to chemotherapy, even in the presence of MSCs. Our results indicate that the mechanism of treatment-induced cell death is late apoptosis as opposed to early apoptosis or necrosis. In this report, we identified stroma-induced ALDH2 as a novel target for treatment of refractory AML in combination with standard chemotherapy.

Our study has some potential limitations. We show promising results regarding ALDH2 inhibition as a strategy to counteract AML chemotherapy resistance; however, we have not validated these findings in vivo. Additionally, the ALDH2 inhibitors tested significantly reduced, but did not completely inhibit, ALDH activity in AML cells. It remains unclear whether other ALDH isoforms contribute to ALDH activity or the ALDH2 inhibitors we used are not sufficiently potent. Moreover, there could be other potential mechanisms of stem-cell phenotype acquisition in AML beyond the scope of our study. Sensitivity to chemotherapy is clearly dependent on Myelodysplastic syndromes-leukemia interactions mediated by chemokines such as the SDF-1/CXCR4 axis [40–42], integrins, and E-selectin-mediated adhesion of AML

cells to vascular cells [43]. Despite the above limitations, our work constitutes an important contribution to the field of AML-microenvironment interactions as it provides a clear mechanistic framework for a better understanding of the role of stromal cells in AML chemotherapy resistance and identifies several potential clinically relevant therapeutic targets.

Our findings set the stage for future studies targeting mediators of AML-stroma interactions. In particular, the effects of ALDH2 inhibitors in combination with chemotherapy and targeted therapies appear worth exploring.

In summary, BM stromal cells induce ALDH activity in AML cells through increased expression of the ALDH2 isoform. BM-MSCs secrete TGF-β1, which exerts its effect through a non-canonical/p38-dependent signaling mechanism, leading to a stem-like phenotype in AML cells. Inhibition of downstream targets of this pathway, such as p38 MAPK, inhibits ALDH activity in AML cells. ALDH2 inhibition sensitizes AML cells to standard chemotherapy in vitro, providing a potential novel therapeutic strategy to overcome AML chemotherapy resistance.

## Supporting information

**S1 Fig. Expression levels of ALDH1L2 and ALDH2 isoforms in TCGA AML dataset. (A)** Data extracted from TCGA AML dataset showing mRNA expression levels of ALD1L2 isoform in 8% of AML cases. **(B)** Data extracted from TCGA AML dataset showing expression of ALDH2 isoform in 12% of AML cases. Figure generated through cBioportal for cancer genomics data analysis tool.
(TIF)

**S2 Fig. TGF-β1 knockdown efficacy of 5 shRNA constructs.** Lentiviral-mediated shRNA was used for stable knockdown of TGF-β1 in human-derived MSCs. TGF-β1 mRNA knockdown efficacy in each shRNA construct was assessed using q-PCR in comparison to control BM-MSCs.
(TIF)

**S3 Fig. Effect of ALDH2 inhibition on ALDH+ AML cells. (A)** OCI-AML3 cells were cultured with or without BM-MSCs for 3 days. Cells were stained with ALDEFLUOR® and ALDH activity was measured by flow cytometry. Diadzin (5 μM) was added to OCI-AML3 cells alone or co-cultured with BM-MSCs, and ALDH activity was compared between treated and untreated cells. **(B)** The same experiment was performed as in **A**, using CVT-10216 instead of diadzin. Dot plots shown here were used to generate bar graphs in Fig 5.
(TIF)

**S4 Fig. ALDH mean fluorescence intensity plots corresponding to the experiments shown in Fig 6A and 6B.**
(TIF)

**S1 Table. Western blot antibodies used in this study.** List of antibodies used for Western blot analysis of protein expression of downstream targets of the TGF-β signaling pathways.
(TIF)

**S2 Table. Taqman PCR primers used in this study.** List of primers used to analyze RNA expression of different ALDH isoforms as well as other TGF-β downstream targets by RT-PCR.
(TIF)

**S3 Table. Clinical information corresponding to 8 AML patient samples used in this study.** Patient demographics and clinical parameters corresponding to the 8 AML peripheral blood

patient samples.
(TIF)

**S1 Raw images.**
(PDF)

## Author Contributions

**Conceptualization:** Venkata Lokesh Battula.

**Formal analysis:** Fouad El Dana, Yuanqing Yan, Venkata Lokesh Battula.

**Funding acquisition:** Michael Andreeff, Venkata Lokesh Battula.

**Investigation:** Bin Yuan, Fouad El Dana, Stanley Ly, Yuanqing Yan, Venkata Lokesh Battula.

**Project administration:** Vivian Ruvolo.

**Supervision:** Venkata Lokesh Battula.

**Writing – original draft:** Fouad El Dana, Venkata Lokesh Battula.

**Writing – review & editing:** Elizabeth J. Shpall, Marina Konopleva, Michael Andreeff, Venkata Lokesh Battula.

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
