## [Decision Letter · Decision Letter 0]

6 Oct 2020

PONE-D-20-26977

Bone marrow stromal cells induce an ALDH+ stem cell-like phenotype and enhance therapy resistance in AML through a TGF-β-p38-ALDH2 pathway

PLOS ONE

Dear Dr. Battula,

Thank you for submitting your manuscript to PLOS ONE. After careful consideration, we feel that it has merit but does not fully meet PLOS ONE’s publication criteria as it currently stands. Therefore, we invite you to submit a revised version of the manuscript that addresses the points raised during the review process.

This manuscript is of interest pending some minor alterations that are needed. The Authors must follow all the point highlighted by the two referees and amend the text. In their rebuttal letter they must specify each point and the correct answer to the referees.

We look forward to receiving your revised manuscript.

Kind regards,

Gianpaolo Papaccio, M.D., Ph.D.

Academic Editor

PLOS ONE

Journal Requirements:

2. Please move Figure S1, describing the results of animal experiments, into the main figures.

3. Please provide additional details regarding participant consent. In the ethics statement in the Methods and online submission information, please ensure that you have specified (1) whether consent was informed and (2) what type you obtained (for instance, written or verbal, and if verbal, how it was documented and witnessed). If your study included minors, state whether you obtained consent from parents or guardians.

4. Please provide additional information about the HEK293T cell line used in this work, including source, history, culture conditions and any quality control testing procedures (authentication, characterisation, and mycoplasma testing). For more information, please see http://journals.plos.org/plosone/s/submission-guidelines#loc-cell-lines.

5. In your Methods section, please provide additional information about the participant recruitment method for the isolation of human MSCs from healthy patient donors. Please ensure you have provided sufficient details to replicate the analyses such as: a) the recruitment date range (month and year), b) a table of relevant demographic details, c) a description of how participants were recruited, and d) descriptions of where participants were recruited and where the research took place.

6. We note that you have indicated that data from this study are available upon request. PLOS only allows data to be available upon request if there are legal or ethical restrictions on sharing data publicly. For more information on unacceptable data access restrictions, please see http://journals.plos.org/plosone/s/data-availability#loc-unacceptable-data-access-restrictions.

7. PLOS ONE now requires that authors provide the original uncropped and unadjusted images underlying all blot or gel results reported in a submission’s figures or Supporting Information files. This policy and the journal’s other requirements for blot/gel reporting and figure preparation are described in detail at https://journals.plos.org/plosone/s/figures#loc-blot-and-gel-reporting-requirements and https://journals.plos.org/plosone/s/figures#loc-preparing-figures-from-image-files. When you submit your revised manuscript, please ensure that your figures adhere fully to these guidelines and provide the original underlying images for all blot or gel data reported in your submission. See the following link for instructions on providing the original image data: https://journals.plos.org/plosone/s/figures#loc-original-images-for-blots-and-gels.

8. Your ethics statement should only appear in the Methods section of your manuscript. If your ethics statement is written in any section besides the Methods, please delete it from any other section.

Reviewers' comments:

Reviewer's Responses to Questions

**Comments to the Author**

1. Is the manuscript technically sound, and do the data support the conclusions?

Reviewer #1: Yes

Reviewer #2: Yes

2. Has the statistical analysis been performed appropriately and rigorously? 

Reviewer #1: Yes

Reviewer #2: Yes

3. Have the authors made all data underlying the findings in their manuscript fully available?

Reviewer #1: Yes

Reviewer #2: Yes

4. Is the manuscript presented in an intelligible fashion and written in standard English?

Reviewer #1: No

Reviewer #2: Yes

5. Review Comments to the Author

Reviewer #1: In this study, the Authors investigated the effect of BM stromal cells on AML cells, signaling pathways activated, and therapeutic targets that contribute to chemotherapy resistance in AML cells. They identified specific inhibitors that could be used in combination with standard chemotherapy for treatment of AML patients.

This manuscript is interesting and focus on new therapeutic targets for AML treatment.

Although this, there are few points that need to be clarified. The Abstract Section must be revised. It must more organic. Regarding apoptosis evaluation, the authors must better explain the results obtained. Did the cells die for necrosis, apoptosis (early o late apoptosis)?

The figure 4D must be discuss in Discussion section. Its explanation is reported only in figure legend. Simplified schematic representation of the interaction between MSCs and AML cells must be fully explained in this section.

The English language must be revised.

Reviewer #2: In this study Authors evaluated the effect of BM-MSCs stroma on AML cells and identified the molecular pathway underlying the induction to stemness and chemoresistance.

The study is interesting and the experiments are well conducted.

Authors have to rewrite abstract and discussion section. Abstract should better explain the aim of the study. In discussion section Authors should enlarge the discussion of the different experiments done.

6. PLOS authors have the option to publish the peer review history of their article (what does this mean?). If published, this will include your full peer review and any attached files.

Reviewer #1: No

Reviewer #2: No

---

## [Author Response · Author response to Decision Letter 0]

26 Oct 2020

Section of Molecular Hematology and Therapy, Department of Leukemia

The University of Texas MD Anderson Cancer Center

1515 Holcombe Blvd Houston Texas, 77030

Joerg Heber, Ph.D. 

Editor-in-Chief, PLoSOne 

October 23, 2020

Dear Dr. Heber,

I am writing to you in reference to our manuscript entitled “Bone marrow stromal cells induce an ALDH+ stem cell-like phenotype and enhance therapy resistance in AML through a TGF-β-p38-ALDH2 pathway” (reference number: PONE-D-20-26977), which was recently peer-reviewed. 

First and foremost, I would like to thank you and the reviewers, on behalf of all authors of the manuscript, for the time and effort that you spent reviewing our work. Your comments have been well received and greatly appreciated, as we believe every single point raised contributes towards improving the quality of our manuscript. I have thoroughly discussed all points with my co-authors and we consequently made appropriate adjustments to our manuscript to address the reviewers’ concerns. We believe our revised manuscript now meets PLoSOne publication criteria and requirements.

Please proceed to the following pages of this letter, where we provide a detailed point-by-point response to the comments and concerns that were raised, highlighting adjustments that were made to address them.

Once again, thank you for considering our work for publication at your esteemed journal.

Sincerely,

Venkata Lokesh Battula, Ph.D.

Response to comments

Academic editor comments:

We have carefully reviewed PLOS ONE manuscript formatting guidelines and used the provided templates to ensure compliance with all submission guidelines. Font changes, formatting, and title page adjustments were made accordingly. Additionally, we have changed the figures layout, separated each figure into a separate file, and converted all files into tiff format. We renamed all submitted files according to journal requirements. PACE tool was used to ensure adherence of our figures to the recommended guidelines

2. Please move Figure S1, describing the results of animal experiments, into the main figures.

We reorganized our figures to an arrangement that includes 6 main figures instead of the previous 4 figures. Figures that include important findings, such as figure S1 describing results of animal experiments, have been moved from supplementary data to main figures. Please note that what was previously figure S1 has now been moved to figure 1A-B. Accordingly, we made the needed changes to figure captions as well as in-text citations to match the new figure arrangement.

3. Please provide additional details regarding participant consent. In the ethics statement in the Methods and online submission information, please ensure that you have specified (1) whether consent was informed and (2) what type you obtained (for instance, written or verbal, and if verbal, how it was documented and witnessed). If your study included minors, state whether you obtained consent from parents or guardians.

In our study, written informed consent was obtained from all healthy individual human MSC donors according to the Declaration of Helsinki. Our study did not include any minors. This information is now added under ‘ethics approval and consent to participate’ in the methods section of the revised manuscript.

4. Please provide additional information about the HEK293T cell line used in this work, including source, history, culture conditions and any quality control testing procedures (authentication, characterization, and mycoplasma testing).

HEK293T cells were purchased from ATCC® (Cat# CRL-3216). Cells were cultured in Dulbecco’s Modified Eagle’s Medium (DMEM) high glucose formulation purchased from Media Tech, Inc supplemented with 10% Fetal Bovine Serum (FBS) purchased from Gibco, and 1% Penicillin/Streptomycin. HEK293T are tested routinely in our laboratory for mycoplasma contamination every 4-6 months. All tests for mycoplasma contamination turned out negative. Please note that this information has been added to the methods section under ‘Cell culture of primary MSCs, leukemia cell lines, and patient samples’. 

5. In your Methods section, please provide additional information about the participant recruitment method for the isolation of human MSCs from healthy patient donors. Please ensure you have provided sufficient details to replicate the analyses such as: a) the recruitment date range (month and year), b) a table of relevant demographic details, c) a description of how participants were recruited, and d) descriptions of where participants were recruited and where the research took place.

Human MSC donors were recruited over a time period between January, 2014 and December, 2019. All donors were healthy individuals between the ages 25-45. All healthy patient donors were recruited according to a protocol approved by MD Anderson Cancer Center institution review board (IRB). Healthy patient donors undergoing bone marrow harvest for allogeneic stem cell transplantation were recruited. All participants provided written informed consent as per the declaration of Helsinki. Participants were recruited and bone marrow samples were collected at the Stem Cell Transplantation and Cellular Therapy Department at MD Anderson Cancer Center. 

6. We note that you have indicated that data from this study are available upon request. PLOS only allows data to be available upon request if there are legal or ethical restrictions on sharing data publicly.

b) If there are no restrictions, please upload the minimal anonymized data set necessary to replicate your study findings as either Supporting Information files or to a stable, public repository and provide us with the relevant URLs, DOIs, or accession numbers..

Please note that we have changed our data availability statement to indicate that all data based on which we make our conclusions in this manuscript are available to the public without any restrictions. We have submitted all supplementary data as individual files. There are neither legal nor ethical restrictions on public sharing of data generated in this research work, therefore, prompt a is not applicable to our study.

For prompt b, we have uploaded our RNA sequencing data to support our findings that TGF-β-associated gene signature is activated in AML cells in the presence of MSCs to a publically available database on NCBI. The accession number for our RNA sequencing dataset is: GSE152996 and is accessible to the public at the following URL: https://urldefense.com/v3/__https://www.ncbi.nlm.nih.gov/geo/query/acc.cgi?acc=GSE152996__;!!PfbeBCCAmug!0C2vyQN-voc9eR8fp-MRw64m1VA7U52_H7qqRNtcfY-BluUKDjg3RMAYRcWIeDES0UM$

All other supplementary data have been submitted as individual supporting information files. The aforementioned adjustments have been included in the revised version of our cover letter.

7. PLOS ONE now requires that authors provide the original uncropped and unadjusted images underlying all blot or gel results reported in a submission’s figures or Supporting Information files.

Original uncropped and unadjusted western blot images have been provided as supporting information files. A sentence has been added to our revised cover letter to reflect this change.

8. Your ethics statement should only appear in the Methods section of your manuscript. If your ethics statement is written in any section besides the Methods, please delete it from any other section.

In the revised manuscript, our ethics statement appears only at the end of our methods section. It has been deleted from other sections of the manuscript.

9. Please include captions for your Supporting Information files at the end of your manuscript, and update any in-text citations to match accordingly.

Please note that we have added captions for supporting information at the end of the manuscript under a new section ‘Supporting information’. In-text citations are also updated to match our new figure arrangement.

Reviewers’ Comments:

Reviewer #1:

1. Abstract section must be revised, it must be more organic

Thank you for your comments. Please note that we have re-written the abstract of our manuscript to include more details about our study and its aim. We have reviewed our abstract several times to improve its quality.

2. Regarding apoptosis evaluation, the authors must better explain the results obtained. Did the cells die from necrosis, apoptosis (early or late apoptosis)?

We have evaluated treatment-induced cell death in OCI-AML3 cells by flow cytometry using both Annexin V and DAPI staining. We found that the majority of cells were Annexin V+/ DAPI+ , indicating that the predominant mechanism of treatment-induced cell death in OCI-AML3 cells is late apoptosis as opposed to early apoptosis and/or necrosis. In the revised manuscript, we included a more elaborate explanation of these findings in the results section under ‘ALDH2 inhibitors sensitize AML cells to standard chemotherapy’, in addition to addressing this question in the our revised discussion section

3. The figure 4D must be discuss in Discussion section. Its explanation is reported only in figure legend. Simplified schematic representation of the interaction between MSCs and AML cells must be fully explained in this section.

The revised discussion section in our manuscript now includes a paragraph which extensively discusses the findings portrayed in our simplified schematic diagram. What was previously figure 4D is now figure 6D in the new figure arrangement. Reference to this figure can be found at the end of our results section as well as the summary statement of our revised discussion.

4. The English language must be revised.

Our manuscript has been subjected to multiple rounds of editing for sentence structure, grammatical errors, typing mistakes, and overall flow of writing. Both corresponding authors read and edited the manuscript multiple times until we wrote our final version of the manuscript. The final version was then sent to all co-authors who also read the manuscript and suggested edits. Additionally, at MD Anderson Cancer Center, there is a specialized department for scientific editing whereby expert scientific writers provide comments and advice to improve manuscripts prior to journal submission. We sent the final version of our manuscript to that department. We received their feedback and implemented all their suggested changes prior to submitting our work to PLOS ONE.

Reviewer #2:

1. The authors should re-write the abstract section. Abstract should better explain the aim of the study.

Thank you for your comments. We have reviewed our abstract and we re-wrote it to include sufficient details and explanations pertaining to the aim of our study.

2. The authors should re-write the discussion section. In the discussion section, the authors should enlarge the discussion of the different experiments that were done.

Please note that we have extensively revised our discussion section, we expanded the discussion to include sections that address specific experiments described in the results section of our manuscript. We also broadened the scope of the discussion and included additional references.

---

## [Decision Letter · Decision Letter 1]

10 Nov 2020

Bone marrow stromal cells induce an ALDH+ stem cell-like phenotype and enhance therapy resistance in AML through a TGF-β-p38-ALDH2 pathway

PONE-D-20-26977R1

Dear Dr. Battula,

We’re pleased to inform you that your manuscript has been judged scientifically suitable for publication and will be formally accepted for publication once it meets all outstanding technical requirements.

Kind regards,

Gianpaolo Papaccio, M.D., Ph.D.

Academic Editor

PLOS ONE

Additional Editor Comments (optional):

Reviewers' comments:

Reviewer's Responses to Questions

**Comments to the Author**

1. If the authors have adequately addressed your comments raised in a previous round of review and you feel that this manuscript is now acceptable for publication, you may indicate that here to bypass the “Comments to the Author” section, enter your conflict of interest statement in the “Confidential to Editor” section, and submit your "Accept" recommendation.

Reviewer #1: All comments have been addressed

Reviewer #2: All comments have been addressed

2. Is the manuscript technically sound, and do the data support the conclusions?

Reviewer #1: Yes

Reviewer #2: (No Response)

3. Has the statistical analysis been performed appropriately and rigorously? 

Reviewer #1: Yes

Reviewer #2: (No Response)

4. Have the authors made all data underlying the findings in their manuscript fully available?

Reviewer #1: Yes

Reviewer #2: (No Response)

5. Is the manuscript presented in an intelligible fashion and written in standard English?

Reviewer #1: Yes

Reviewer #2: (No Response)

6. Review Comments to the Author

Reviewer #1: (No Response)

Reviewer #2: (No Response)

7. PLOS authors have the option to publish the peer review history of their article (what does this mean?). If published, this will include your full peer review and any attached files.

Reviewer #1: No

Reviewer #2: No

---

## [Editor Report · Acceptance letter]

17 Nov 2020

PONE-D-20-26977R1 

Bone marrow stromal cells induce an ALDH^+^ stem cell-like phenotype and enhance therapy resistance in AML through a TGF-β-p38-ALDH2 pathway 

Dear Dr. Battula:

I'm pleased to inform you that your manuscript has been deemed suitable for publication in PLOS ONE. Congratulations! Your manuscript is now with our production department. 

Kind regards, 

on behalf of

Prof. Gianpaolo Papaccio 

Academic Editor

PLOS ONE